

# Hepatic autotaxin overexpression in infants with biliary atresia

Wanvisa Udomsinprasert[1,2], Paisarn Vejchapipat[3], Naruemon Klaikeaw[4], Voranush Chongsrisawat[5], Yong Poovorawan[5] and Sittisak Honsawek[1]

[1] Osteoarthritis and Musculoskeleton Research Unit, Department of Biochemistry, Faculty of Medicine, King Chulalongkorn Memorial Hospital, Thai Red Cross Society, Chulalongkorn University, Bangkok, Thailand
[2] Department of Biochemistry, Faculty of Pharmacy, Mahidol University, Bangkok, Thailand
[3] Department of Surgery, Faculty of Medicine, King Chulalongkorn Memorial Hospital, Thai Red Cross Society, Chulalongkorn University, Bangkok, Thailand
[4] Department of Pathology, Faculty of Medicine, King Chulalongkorn Memorial Hospital, Thai Red Cross Society, Chulalongkorn University, Bangkok, Thailand
[5] Center of Excellence in Clinical Virology, Department of Pediatrics, Faculty of Medicine, King Chulalongkorn Memorial Hospital, Chulalongkorn University, Bangkok, Thailand

Corresponding author
Sittisak Honsawek,
sittisak.h@chula.ac.th

## ABSTRACT

**Background**. Autotaxin (ATX) is a secreted glycoprotein that is involved in the development of hepatic fibrogenesis via the enzymatic production of lysophosphatidic acid. The aim of this study was to investigate hepatic expression of ATX in biliary atresia (BA) compared with non-BA liver controls and to examine the association between ATX expression and clinical outcome in BA.

**Methods**. Liver specimens from BA infants ($n = 20$) were compared with samples from infants who underwent liver biopsy for reasons other than BA ($n = 14$) and served as controls. Relative mRNA and protein expression of ATX were quantified using real-time polymerase chain reaction (PCR) and immunohistochemistry. Masson's Trichrome staining was performed to determine the degree of liver fibrosis.

**Results**. Quantitative real-time PCR demonstrated overexpression of *ATX* mRNA in BA livers. In immunohistochemical evaluation, ATX was positively stained on the hepatic parenchyma and the biliary epithelium in BA patients, as compared to non-BA controls. The immunostaining score of ATX in BA livers was also significantly higher than that observed in non-BA livers ($P < 0.001$). Subgroup analysis revealed that ATX expression in the patients with poor outcomes was significantly greater than in those with good outcomes ($P = 0.03$). Additionally, there was a positive correlation between hepatic ATX expression and Metavir fibrosis stage in BA livers ($r = 0.79$, $P < 0.001$).

**Discussion**. This study found that mRNA and protein expression of ATX were increased in BA livers. High hepatic ATX expression at the time of Kasai operation was associated with liver fibrosis and outcome in BA, suggesting that ATX may serve a role as a promising biomarker of the prognosis in biliary atresia.

## INTRODUCTION

Biliary atresia (BA) is a neonatal cholestasis disease that is characterized by fibrosclerosing and inflammatory obliteration of the biliary tracts, which leads to progressive liver damage (*Hartley, Davenport & Kelly, 2009*). Kasai portoenterostomy (KPE), the primary treatment for BA, establishes good bile flow and facilitates long-term survival. However even after timely KPE, a number of infants are at risk of developing new biliary obstruction that could lead to chronic cholestasis, increased fibrosis, cirrhosis, and eventually to end-stage liver disease. As such, BA is the leading cause of liver transplantation in children. Although the precise pathogenesis of BA remains elusive, possible etiologies include viral infection, toxins, chronic inflammatory or immune-mediated bile duct injury, and abnormalities in bile duct development (*Bezerra, 2005*). Increased understanding of what causes inflammatory cholangiopathy in BA could lead to therapies aimed at protecting the intrahepatic biliary system from inflammation-mediated fibrosis. However, the molecular mechanisms involved in the pathogenesis of liver fibrosis in BA have not yet been fully and clearly established. Hepatic fibrosis is a reversible physiologic and pathologic event, and the possible role of cytokine-mediated pathogenesis in this disorder is of great interest to many researchers.

Autotaxin [ATX; ectonucleotide pyrophosphatase/phosphodiesterase family member 2 (ENPP2)] is a secreted lysophospholipase D that generates the lipid mediator lysophosphatidic acid (LPA) from extracellular lysophospholipids—predominantly from lysophosphatidylcholine (*Tokumura et al., 2002*). ATX-LPA signaling has been implicated in multiple biological and pathophysiological processes, including vasculogenesis, cholestatic pruritus, tumor progression, and fibrosis via six distinct G-protein-coupled LPA receptors (LPAR1-6) (*Umezu-Goto et al., 2002*). *Ikeda et al. (2003)* identified a potential link between the ATX-LPA axis and liver fibrosis when they found that intradermal LPA induces hepatic stellate cell (HSC) proliferation, stimulates their contraction, and inhibits their apoptosis. HSCs are known as prototypic profibrogenic cells in the hepatic parenchyma. After transformation into myofibroblasts in response to a liver injury, HSCs start to produce abundant extracellular matrices and profibrogenic cytokines, such as ATX-derived LPA. In addition, both LPA and ATX concentrations were increased in chronic hepatitis C patients with liver fibrosis (*Watanabe et al., 2007*). The *ATX-LPA* axis has also been reported to be up-regulated in human hepatocellular carcinoma (*Park et al., 2011*; *Wu et al., 2010*), thereby establishing the possible influence of ATX in inflammation-related hepatic fibrosis disorders like biliary atresia.

Although circulating ATX levels have been shown to be associated with liver fibrosis, there is limited information on ATX expression in liver tissue and regarding the association between ATX expression and BA outcomes. Accordingly, the aims of this study were to investigate mRNA and protein expression of ATX in liver tissues from BA patients compared with non-BA controls and to evaluate whether hepatic ATX expression is associated with outcome parameters in BA infants.

## MATERIALS AND METHODS

### Patients and liver specimens

The study protocol conformed to the ethical standards outlined in the Declaration of Helsinki and was approved by the Institutional Review Board of the Faculty of Medicine, Chulalongkorn University (IRB No. 549/57). All parents of children were fully informed regarding the study protocol and procedures prior to the children entering the study. Written informed consent was obtained from the participants' parents.

Perioperative liver biopsies were obtained from 20 BA infants at the time of KPE and 14 non-BA patients at the Department of Surgery, King Chulalongkorn Memorial Hospital during the July 2005 to July 2007 study period. All BA infants were invited to participate in this study based on the following criteria: (1) diagnosis of type 3 (uncorrectable) isolated BA and they underwent Kasai procedure; (2) availability of clinical details and long term follow-up after surgery; and (3) availability of archived glass slides or paraffin blocks of wedge liver specimens taken at KPE. Infants diagnosed with BA or non-BA were included based on clinical, cholangiographic, and histologic findings. Non-BA patients with no history of immune-mediated diseases served as controls. Non-BA control samples were collected from six patients with choledochal cyst, four patients with thalassemia, three patients with neuroblastoma, and one patient with hepatoblastoma. Liver biopsies taken from non-BA controls were obtained during procedures that were required for medical reasons. Liver specimens from age-matched healthy controls could not be obtained due to ethical concerns about harvesting liver tissue from healthy infants.

Demographic and clinical data collected at the time of KPE included age, albumin, total bilirubin (TB), and alanine aminotransferase (ALT). Laboratory investigations were performed on a Roche Hitachi 912 chemistry analyzer (Roche Diagnostics, Basel, Switzerland). In order to associate ATX hepatic expression with outcome at six months post-Kasai in BA, the infants were divided into good outcome and poor outcome based on their levels of serum TB, ALT, and clinical findings. Nine patients with good outcomes had good bile flow after KPE. The stool color turned from pale to yellowish for six months following successful surgery. The serum TB returned to normal with satisfactory liver function (TB < 2 mg/dL, ALT < 100 IU/L). Another 11 patients with poor outcome had cholestasis after six months KPE and severe liver dysfunction (TB $\geq$ 2 mg/dL, ALT $\geq$ 100 IU/L).

### RNA extraction and quantitative real time-PCR for mRNA expression of ATX

Of 20 BA infants and 14 non-BA controls, 15 BA livers and five non-BA liver specimens (choledochal cyst) were snap-frozen in liquid nitrogen-cooled isopentane and stored at −80 °C and only available for *ATX* mRNA expression. Total RNA was isolated from liver biopsies using RNeasy Mini Kit (Qiagen, Hilden, Germany) with cDNA that was reverse transcribed using TaqMan Reverse Transcription Reagents (Applied Biosystems, Inc., Foster City, CA, USA). Real-time PCR was performed using QPCR Green Master Mix HRox (biotechrabbit GmbH, Hennigsdorf, Germany) on StepOnePlus Real-Time PCR System (Applied Biosystems, Inc., Foster City, CA, USA). Primers used for *ATX*

and glyceraldehyde 3-phosphate dehydrogenase (*GADPH*) amplification were, as follows: *ATX* forward primer 5′-CGTGGCTGGGAGTGTACTAA-3′; *ATX* reverse primer 5′-AGAGTGTGTGCCACAAGACC-3′, as previously described (*Kondo et al., 2014*); *GADPH* forward primer 5′-GTGAAGGTCGGAGTCAACGG-3′; and, *GADPH* reverse primer 5′-TCAATGAAGGGGTCATTGATGG-3′. Real-time PCR was performed, as follows: (initial step) 95 °C for 10 min, followed by 40 cycles of 95 °C for 15 sec, and then 60 °C for 1 min. Relative mRNA expression of *ATX* was normalized to *GADPH* as an internal control and was determined using $2^{-\Delta\Delta Ct}$ method.

## Masson's Trichrome staining

Masson's Trichrome staining was conducted according to the manufacturer's protocol (Genmed Scientifics, Wilmington, DE, USA). The collagen fiber was stained blue, the nuclei were stained black, and the background was stained red. Liver fibrosis was evaluated according to the Metavir grading system (*Bedossa & Poynard, 1996*) as follows: F0, no fibrosis; F1, mild fibrosis in the portal area; F2, mild bridging fibrosis in the adjacent portal area; F3, severe bridging fibrosis in the adjacent portal area; and F4, cirrhosis and annular fibrosis with nodule formation.

## Immunohistochemical analysis for protein expression of ATX

All liver specimens of 20 BA patients and 14 non-BA controls were paraffin-embedded and then sectioned according to standard protocols. Routine staining with hematoxylin and eosin, and immunohistochemical staining with antibodies was performed to detect protein expression of ATX (Merck Millipore, Darmstadt, Germany). For ATX staining, cells with brown stained cytoplasm were scored as positive. All tissue sections were analyzed by a pathologist who was blinded to patient clinical status and diagnosis. Immunoreactivity of ATX in the biliary epithelium and the parenchyma was semi-quantitatively analyzed for percentage of positive cells and intensity of staining. A percentage of positive cells <1% was scored as 0; 1%-25% as 1; >25%–50% as 2; >50%–75% as 3; and, >75% as 4. Intensity of ATX immunostaining was determined using the following staining scores: 0, no staining; 1, weak staining; 2, moderate staining; and 3, strong staining. Final results were scored by the total score [total scores = ((score of positive cell+score of intensity) × 100)/maximum score of both parameters]. Using the aforementioned staining scores, the positive areas of ATX positive cells were determined by measuring five randomly selected microscopic fields (400×) on each slide.

## Statistical analysis

All statistical analyses were performed using SPSS Statistics version 22.0 (SPSS, Inc., Chicago, IL, USA). Demographic and clinical characteristics between groups were evaluated using Chi-square tests and unpaired Student's $t$-tests where appropriate. The comparisons of ATX expression between groups were performed by Mann–Whitney $U$-tests. Correlations were analyzed by Spearman's rank correlation. The curves for survival were drawn according to the Kaplan–Meier analysis with end points of death. The differences of survival curves were determined using log-rank test. Data are presented

**Table 1  Demographic and clinical characteristics of BA patients and non-BA controls.**

| Characteristics | BA patients (n = 20) | Non-BA controls (n = 14) | P-value |
|---|---|---|---|
| Age (days) | 91.1 ± 7.0 | 897.5 ± 315.2 | <0.01 |
| Gender (female:male) | 12:8 | 8:6 | 0.6 |
| Albumin (g/dL) | 4.1 ± 0.1 | 4.2 ± 0.3 | 0.3 |
| Total bilirubin (mg/dL) | 12.2 ± 0.8 | NA | – |
| ALT (IU/L) | 191.8 ± 25.8 | NA | – |
| Hepatic ATX expression (%) | 50.0 ± 5.9 | 13.4 ± 5.0 | <0.001 |

**Notes.**

Data presented as mean ± SEM.

P-value < 0.05 indicates a statistically significant difference in clinical data between BA patients and non-BA controls at the time of Kasai portoenterostomy (KPE).

Abbreviations: BA, biliary atresia; ALT, alanine aminotransferase; ATX, autotaxin; NA, Data not available; SEM, standard error of the mean.

as mean ± standard error of the mean (SEM). A P-value < 0.05 was considered to be statistically significant for differences and correlations.

# RESULTS

## Clinical characteristics of study participants

Baseline characteristics of BA infants and non-BA controls are summarized in Table 1. There was no statistically significant difference in gender ratio between BA patients and non-BA controls. All non-BA participants had no clinical jaundice. The diagnosis of non-BA subjects included six choledochal cysts, four thalassemias, three neuroblastomas, and one hepatoblastoma.

## Relative mRNA expression of ATX

To identify mRNA expression of *ATX* in infants with BA, relative *ATX* mRNA expression was quantified by real-time polymerase chain reaction (PCR) in liver biopsies from BA patients (n = 15) and non-BA controls (n = 5). Relative *ATX* mRNA expression was found to be significantly higher in BA livers than non-BA liver controls (P < 0.05) (Figs. 1A; 1B).

## Immunohistochemistry analysis of ATX protein expression

Immunohistochemical evaluation for ATX protein expression was performed in both BA and non-BA liver tissues. Representative immunohistochemical findings of ATX are illustrated in Fig. 2. In congenital BA liver specimens, overexpression of ATX was detectable in the hepatic parenchyma, biliary epithelial cells, and cells of the surrounding connective tissue. In contrast, ATX expression was only scarcely evident in non-BA control livers, being demonstrated as faint cytoplasmic staining (Fig. 2A). The distribution of ATX in positive cells was classified as cytoplasm-localized pattern.

In order to compare expression levels of ATX protein between BA patients and non-BA controls, staining intensity and percentage of ATX positive cells were assessed by visual scoring method. In BA livers at the time of KPE, hepatic ATX protein expression was significantly higher than that in non-BA controls when measured by the total score of

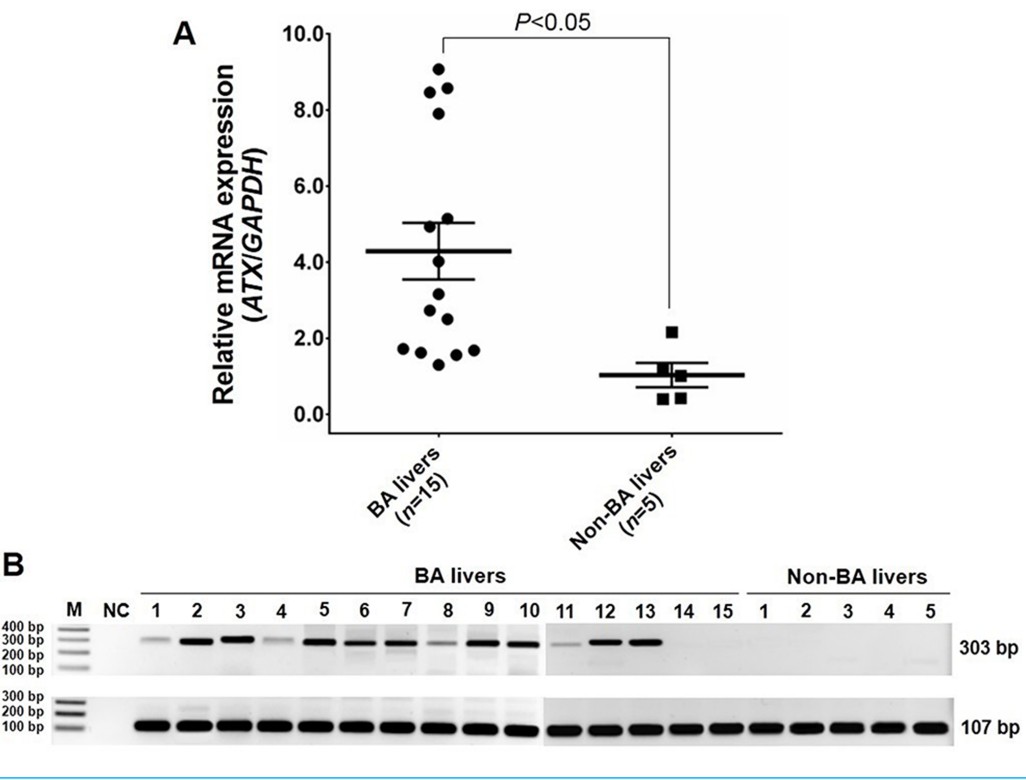

**Figure 1   Relative *ATX* mRNA expression between BA livers and non-BA liver controls.** (A) Up-regulated mRNA expression of *ATX* normalized by *GAPDH* in livers from BA infants. (B) Representative gel of *ATX* and *GAPDH* products from real-time PCR analysis. Abbreviations: M, molecular weight marker; NC, non-template control.

staining on histologic liver sections ($P < 0.001$) (Fig. 3A). There was no association between ATX mRNA and protein expression in BA livers.

## Hepatic ATX protein expression in BA subgroups

To determine whether hepatic ATX protein expression would be associated with poor outcomes in BA patients, we classified BA children according serum TB, liver enzymes, and clinical findings at six months post-Kasai into patients with poor outcome ($n = 11$) and patients with good outcome ($n = 9$). Table 2 demonstrates the clinical characteristics of the BA subgroups based on clinical outcome at six months post-operation. Subsequent analysis demonstrated the mean immunoreactive score of ATX protein expression in BA patients with poor outcome was significantly greater than in patients with good outcome ($P = 0.03$) (Fig. 3B). We further analyzed the correlation between hepatic ATX protein expression and markers of liver function in BA patients. The results showed no association between hepatic ATX protein expression and liver function parameters in BA infants at the time of KPE.

## Increased hepatic ATX protein expression and liver fibrosis

The portal areas illustrated various degrees of fibrosis, but fibrous septa were generally broad with lobular extension (Fig. 4A). The liver fibrosis grade was F0 in five cases, F1

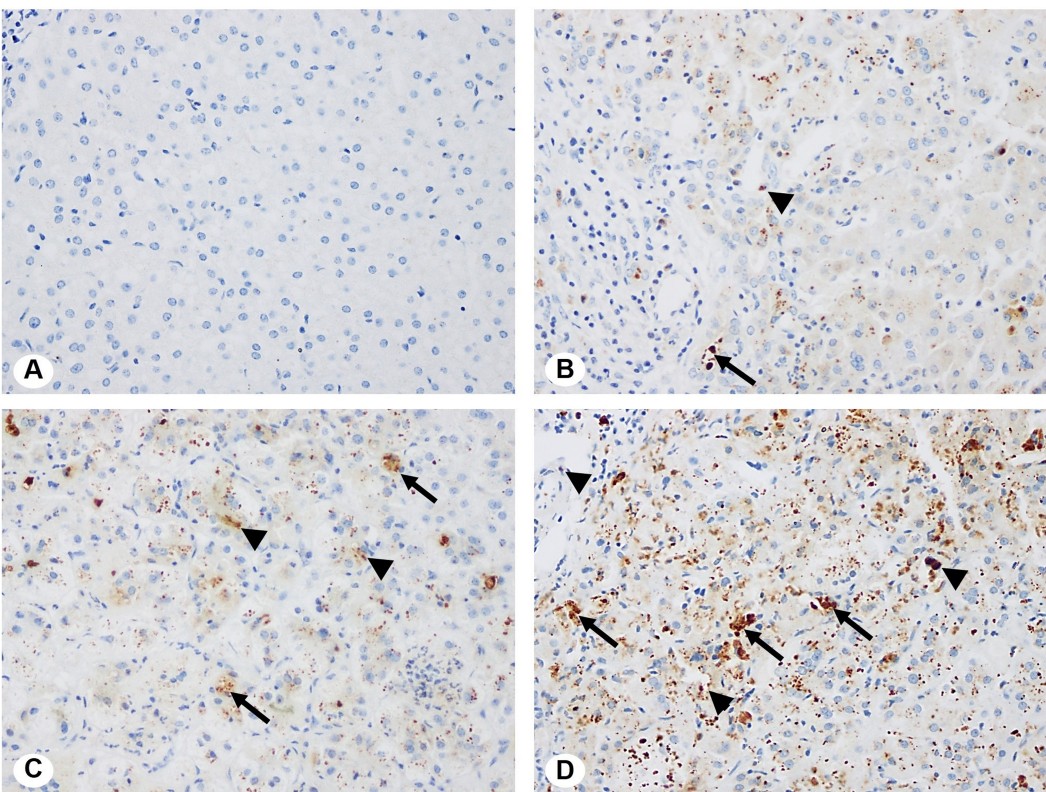

**Figure 2** **Immunohistochemical staining for ATX protein expression.** Specific staining of ATX protein is represented by brown coloration. Expression of ATX in BA livers was observed mostly in the hepatic parenchyma (arrows) and biliary epithelium (arrowheads). ATX staining scores were defined, as follows. (A) 0 = no expression of ATX in a liver used as a control. (B) 1 = mild expression of ATX in BA liver. (C) 2 = moderate expression of ATX in BA liver. (D) 3 = strong expression of ATX in BA liver. (Original magnifications 400×).

**Table 2** **Demographic and clinical characteristics of BA patients based on clinical outcome at six months post-Kasai.**

| Characteristics | BA patients with poor outcome ($n = 11$) | BA patients with good outcome ($n = 9$) | *P*-value |
|---|---|---|---|
| Age at operation (days) | $98.2 \pm 11.3$ | $77.0 \pm 4.9$ | 0.1 |
| Gender (female:male) | 6:5 | 6:3 | 0.4 |
| Albumin (g/dL) | $3.5 \pm 0.2$ | $4.1 \pm 0.1$ | <0.05 |
| Total bilirubin (mg/dL) | $5.7 \pm 1.7$ | $0.4 \pm 0.1$ | <0.001 |
| ALT (IU/L) | $140.2 \pm 27.2$ | $82.8 \pm 16.2$ | 0.01 |
| Hepatic ATX expression (%) | $61.4 \pm 5.9$ | $36.1 \pm 9.4$ | 0.03 |

**Notes.**
Data presented as mean $\pm$ SEM.
*P*-value $< 0.05$ indicates a statistically significant difference in clinical data between BA patients with poor outcome and good outcome at 6 months post-Kasai.
Abbreviations: BA, biliary atresia; ALT, alanine aminotransferase; ATX, autotaxin; SEM, standard error of the mean.

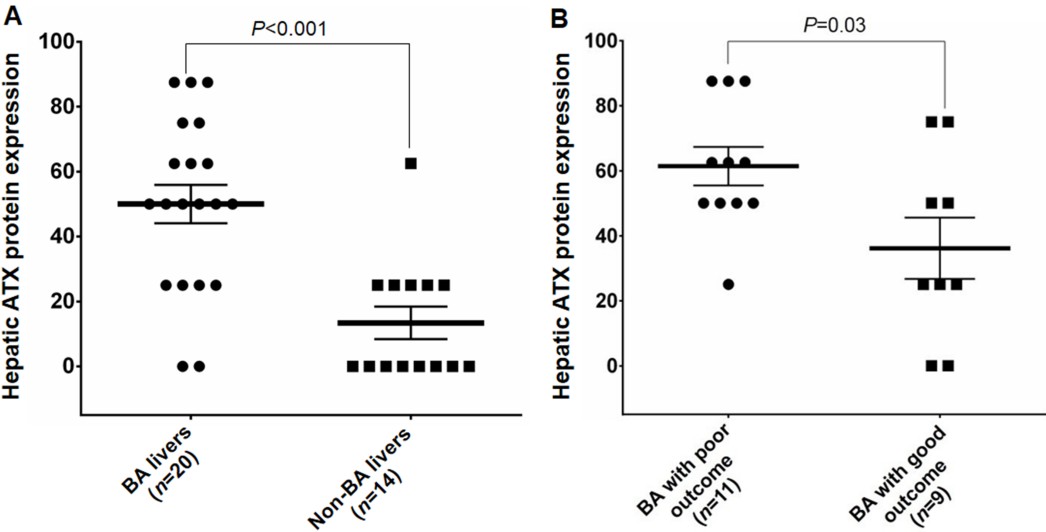

**Figure 3 Hepatic ATX protein expression in study subjects between different groups.** (A) Hepatic ATX expression in BA patients and non-BA patients. (B) Hepatic ATX expression in BA patients with poor outcomes and good outcomes.

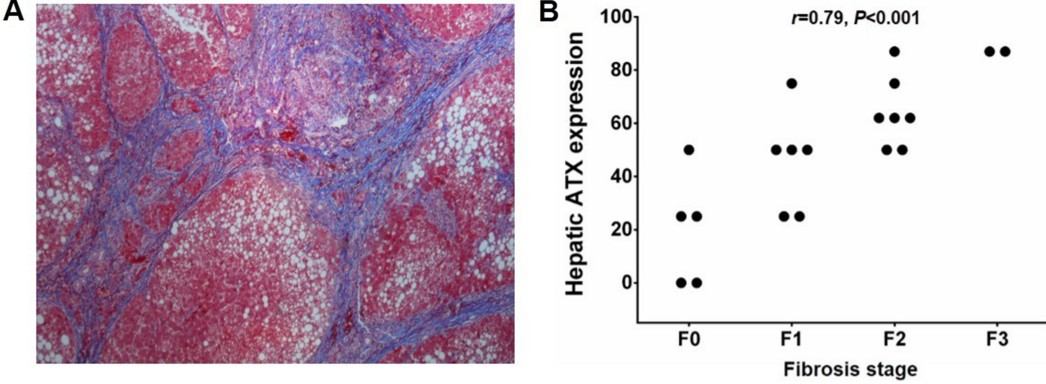

**Figure 4 Hepatic ATX expression and liver fibrosis.** (A) Histopathological analysis of liver samples in BA (Masson Trichrome staining, magnifications 100×). (B) Hepatic ATX protein expression correlated positively with Metavir fibrosis stage in BA ($r = 0.79$, $P < 0.001$).

in six cases, F2 in seven cases, and F3 in two cases. As shown in Fig. 4B, hepatic ATX protein expression was positively correlated with Metavir fibrosis stage in BA; ($r = 0.79$, $P < 0.001$).

## Survival curve analysis

We performed Kaplan–Meier analysis to investigate the overall survival curve of all 20 BA children. The 10-year survival rate with native livers of all BA children were estimated 80%, as shown in Fig. 5A. When stratified into low and high ATX protein expression using the cut-off value of 50%, the overall survival rates at 10 years were 91.7% for those

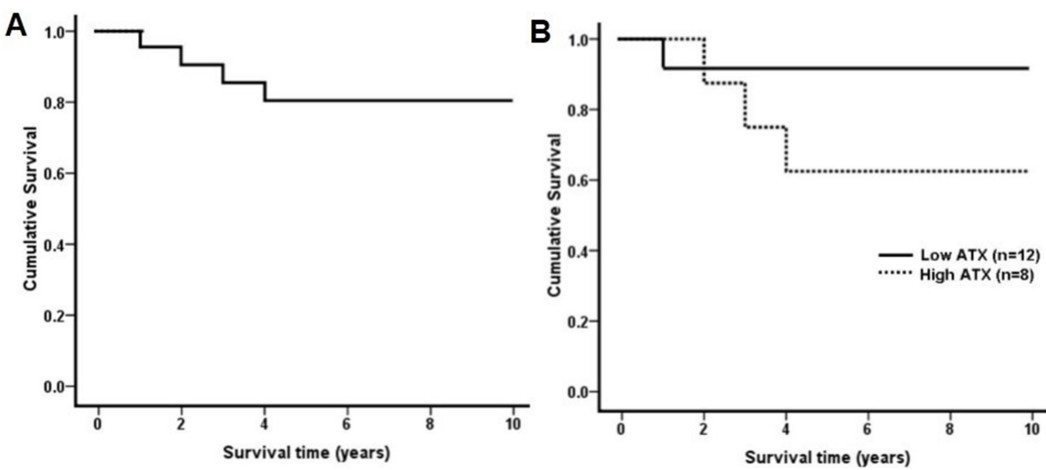

**Figure 5  Kaplan–Meier survival curve of BA children after KPE over 10 years.** (A) The overall survival curve of 20 BA reveals that 10-year survival rates with native livers are 80%. (B) Survival curve comparisons display that BA children with low ATX expression ($n = 12$) have 10-year survival greater than those with high ATX expression ($n = 8$) (log-rank, $X^2 = 2.17$, $P = 0.14$).

with low expression and 62.5% for those with high expression. Survival rate was greater in BA patients with low ATX expression than those with high ATX expression (log-rank, $X^2 = 2.17$, $P = 0.14$) (Fig. 5B).

## DISCUSSION

Despite extensive research efforts, the understanding of mechanisms that regulate biliary atresia (BA) progression following Kasai portoenterostomy (KPE) remains unclear. Biliary atresia causes rapidly progressive liver fibrosis and cirrhosis in neonates, and is the most common indication for liver transplantation in children. Although the precise cause of liver fibrosis in BA remains unclear, several cytokines have been implicated in the regulation of hepatic fibrogenesis (*Kanzler et al., 1999*; *Williams et al., 2000*; *Farrington et al., 2010*; *Xiao et al., 2015*; *Iordanskaia et al., 2015*; *Klemann et al., 2016*). In a previous study, we reported association between elevation of circulating autotaxin (ATX) and poor outcomes in BA patients—especially severity of fibrosis (*Udomsinprasert et al., 2015*). This is important evidence that supports the hypothesis that ATX may serve a role as a potential biomarker of the prognosis in BA. In the present study, we investigated mRNA and protein expression of ATX in liver biopsies from BA infants compared with non-BA controls, and we found up-regulated *ATX* mRNA in BA patients. We then performed immunohistochemical analysis to determine protein expression of ATX and found an intense increase in ATX staining in BA infants, predominantly in the hepatic parenchyma and biliary epithelium at the time of KPE.

The biological outcome of ATX has been shown to induce a variety of inflammatory phenomena via LPA activity, and its role in disease pathophysiology has been verified in several diseases (*Umezu-Goto et al., 2002*), making ATX-derived LPA signaling an attractive therapy. Indeed, emerging evidence suggests that the liver is the main source of

ATX metabolism in both human and animal models (*Ikeda & Yatomi, 2012*). Additionally, ATX expression has been detected on all types of liver cells, including the biliary epithelium (*Kremer et al., 2010*). This suggests the possibility of ATX having a regulatory role in the liver. Our result demonstrated that *ATX* mRNA expression was up-regulated in the livers of BA infants when compared with non-BA liver controls. This finding is consistent with a previous investigation that reported overexpression of *ATX* in liver tissues of patients with hepatocellular carcinoma (HCC) (*Cooper et al., 2007*), suggesting that up-regulated *ATX* expression is associated with hepatic damage and liver fibrosis. In addition to up-regulation of *ATX* mRNA expression in the livers of BA infants, an increase in hepatic protein expression of ATX was also demonstrated. A recently published report also confirmed that overexpression of ATX protein was specifically associated with inflammation and cirrhosis in HCC patients (*Wu et al., 2010*). This was in agreement with our finding, implying that ATX may play a role in inflammation that is related to progressive BA. Furthermore, our immunohistochemistry data demonstrated positive cytoplasmic ATX expression in inflammatory cells and biliary epithelial cells. Indeed, hepatic ATX expression varied between different stages of cholestasis. This study also revealed that there was a positive correlation between hepatic ATX expression and the degree of fibrosis, suggesting the possibility of ATX as a predictive tool for discriminating between good and poor prognosis of clinical outcome in postoperative BA. Consistent with our finding, *Wunsch et al. (2016)* have demonstrated elevated ATX expression in chronic cholestatic diseases. In addition, a recent study revealed that high ATX expression was detected in hepatocellular carcinoma and was correlated with histological grade and survival rate (*Memet et al., 2017*). Rather, ATX might be associated with the nature of BA disease itself. Thus, it is reasonable to postulate that increased expression of ATX in BA livers might reflect a defensive response by the body to fight against hepatic impairment, or may simply be a compensatory response to ATX, which leads to its compensatory up-regulation.

The potential significance of elevated ATX expression in BA remains unclear. The aberrant production of ATX may result in the altered activation of LPA signaling pathways via G-protein-coupled LPA-receptors, and may not be limited to activation of signaling-associated cell proliferation, migration, and apoptosis. Hepatic stellate cells (HSCs) are known to play a major role in the fibrotic process in the liver and they may contribute to the prognosis of BA. For this reason, multiple factors with potentially fibrogenic activities in the liver have been evaluated due to their effects on HSC activation and apoptosis. Regarding the potential effect of ATX-mediated LPA on HSCs, LPA has been shown to stimulate the contractility of HSCs and to inhibit their apoptosis via Rho/Rho kinase activation (*Ikeda et al., 2003*; *Yanase et al., 2003*). Although ATX may not play a primary role in the pathogenesis of liver fibrosis, it may accelerate fibrogenesis by stimulating the proliferation of HSCs in patients with liver fibrosis via its ability to produce LPA. This hypothesis has been supported by the recent observation that specific ATX transgenic overexpression and/or gene disruption from hepatocytes in mice models of chronic liver injury established a liver profibrotic role for ATX/LPA (*Kaffe et al., 2017*). A more recent study by *Bain et al. (2017)* found that a selective ATX inhibitor (PAT-505) markedly reduced liver fibrosis in mouse models. From those findings, we observe that hepatic ATX

expression was associated with an adverse clinical outcome in BA, which lends support to the hypothesis that inhibiting ATX as part of an antifibrotic model could serve as a novel therapeutic approach for treatment of hepatic fibrosis in BA patients. Taken together, the aforementioned findings suggest that the aberrant expression of ATX may be used as a promising biomarker for predicting the progression and prognosis of biliary atresia after Kasai portoenterostomy. Further experiments that isolate biliary epithelial cells and HSCs from BA livers will be required to determine the precise biological and pathological significance of the findings and observations presented in this report.

This study has some mentionable limitations. The most notable limitation is the fact that we were unable to obtain age-matched liver tissue from healthy infants due to ethical considerations. The limited availability of frozen liver biopsies from non-BA controls could have posed significant challenges to the study. Second, the sample size of our study population is relatively small. This is due, in large part, to the fact that BA is a relatively rare disorder. The limited number of subjects makes it challenging to show significant correlations of all parameters in BA patients. Future larger scale, multicenter studies should be conducted to verify our conclusions. Another caveat is the lack of data regarding the circulating ATX levels, total serum bile salt levels, and cholestatic pruritis. We recognize that these could be addressed by prospective longitudinal multicenter cohorts. Further research of costaining on BA liver specimens will identify the cellular fractions expressing ATX. Finally, the causal association between hepatic ATX expression and BA was not fully addressed in the study. Additional research is required to evaluate whether increased hepatic ATX expression is causally related to progressive BA or whether it is simply a compensatory response to the disease.

## CONCLUSIONS

The current study presents evidence of the up-regulation of *ATX* mRNA expression in liver specimens of BA patients, as compared to specimens from livers of non-BA controls. ATX was expressed not only in the hepatic parenchyma, but also in biliary epithelial cells of BA infants at the time of KPE. These findings suggest that ATX expression could be related to liver fibrosis and outcome in biliary atresia. Further investigations examining the possible effect of selective ATX inhibitors on inflammation and progression of liver fibrosis in BA are needed for the development of non-transplant therapeutic strategies to prevent the progression of this devastating disease in affected infants.

## ACKNOWLEDGEMENTS

The authors gratefully acknowledge Center of Excellence in Clinical Virology and Chulalongkorn Medical Research Center (ChulaMRC) for providing research facilities, and Preecha Ruangvejvorachai, Thamonwan Woraruthai, and Dong Zhan for technical assistance. We thank Kevin P. Jones for reviewing and proofreading the manuscript.

### Funding

This work was supported by the Thailand Research Fund (RSA5880019), the Research Chair Grant from the National Science and Technology Development Agency, and Osteoarthritis and Musculoskeleton Research Unit. Wanvisa Udomsinprasert was supported by a grant from the 100th Anniversary Chulalongkorn University Fund for Doctoral Scholarship and Overseas Research Experience Scholarship for Graduate Student. The funders had no role in study design, data collection and analysis, decision to publish, or preparation of the manuscript.

### Grant Disclosures

The following grant information was disclosed by the authors:
Thailand Research Fund: RSA5880019.
National Science and Technology Development Agency.
Osteoarthritis and Musculoskeleton Research Unit.
Chulalongkorn University Fund.

### Competing Interests

The authors declare there are no competing interests.

### Author Contributions

- Wanvisa Udomsinprasert conceived and designed the experiments, performed the experiments, analyzed the data, prepared figures and/or tables, authored or reviewed drafts of the paper, approved the final draft.
- Paisarn Vejchapipat and Sittisak Honsawek conceived and designed the experiments, performed the experiments, analyzed the data, contributed reagents/materials/analysis tools, prepared figures and/or tables, authored or reviewed drafts of the paper, approved the final draft.
- Naruemon Klaikeaw conceived and designed the experiments, performed the experiments, analyzed the data, authored or reviewed drafts of the paper, approved the final draft.
- Voranush Chongsrisawat conceived and designed the experiments, performed the experiments, authored or reviewed drafts of the paper, approved the final draft.
- Yong Poovorawan conceived and designed the experiments, performed the experiments, analyzed the data, contributed reagents/materials/analysis tools, prepared figures and/or tables, authored or reviewed drafts of the paper, approved the final draft, provided samples and clinical data.

### Human Ethics

The following information was supplied relating to ethical approvals (i.e., approving body and any reference numbers):

The study protocol conformed to the ethical standards outlined in the Declaration of Helsinki and was approved by the Institutional Review Board of the Faculty of Medicine, Chulalongkorn University (IRB No. 549/57).

## Data Availability

The raw data are provided in a Supplemental File.

## Supplemental Information

Supplemental information for this article can be found online at http://dx.doi.org/10.7717/peerj.5224#supplemental-information.

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
