# Peer review of "Hepatic autotaxin overexpression in infants with biliary atresia"

_PeerJ, doi:10.7717/peerj.5224_

## Round 0.1 · original submission · Major Revisions

Thank you for the submission to PeerJ. We are inviting you to revise the manuscript to solve all of comments suggested by the two reviewers, especially the comment to expand the control group.

·

Basic reporting

I see a couple of minor issues in the basic reporting:

A. The statement "...infants who underwent liver biopsy for non-BA patients...," found in the abstract (page 2, line 31) needs to be modified. Perhaps the authors meant "...infants who underwent liver biopsy for reasons other than BA."

B. Line 119-121: As written, it sounds like 15 BA patients and 5 non-BA patients had only mRNA samples ("...and only available for ATX mRNA expression"); yet, Figure 3 clearly shows immunostaining results for all 7 non-BA patients. I suggest revising the sentence in the Methods section to make it more clear.

Experimental design

I see at least one major issue with the experimental design:

A. Most of the patients in the non-BA group had choledochal cysts (6 out of 7). Thus, an alternative explanation of the authors' data is that patients with that condition have lower expression of autotaxin in the liver. In fact, there is clearly one patient in the non-BA control group that had higher autotaxin than all of the others; is that patient the one who did NOT have a choledochal cyst? I am aware of the difficulty of obtaining liver biopsy samples from infants, but I strongly suggest that the authors expand their control group. This is especially necessary as the authors mention several other liver diseases that are associated with ATX expression and LPA in the Discussion.

Validity of the findings

Please see my comment for experimental design, as it also pertains to the validity of the findings.

Additional comments

This is a small, but potentially useful study that adds to our mechanistic knowledge of a poorly understood disease. Generally, I'd like to see larger sample sizes, but it is difficult to recruit BA patients prior to or at the time of operation. I would definitely like to see a larger and more diverse control group, though, and that should be doable.

Reviewer 2 ·

Basic reporting

The following is a review of a paper from Udomsinprasert et al. on the presence of autotaxin in patients with bilary atresia. Given the uncommonness of BA, the authors collected a decent number of samples, as well as controls and tried to assess expression levels of autotaxin in BA infants given recent report’s of autotaxins role in fibrogenesis. The authors provide convincing data with solid IHC and well analyzed results. This is an appropriate paper for this journal.

Experimental design

Comments: The authors provide convincing data that ATX may serve a role as a biomarker of prognosis in BA; however, this does not mean that ATX has a role in the disease. Comments indicating ATX’s potential role should be eliminated. The paper should focus on the potential for ATX as a biomarker.

The p value for the 10 year survival curve was .14 which is not significant by most standards. Why did the authors choose 10 year? Why not 5 year where there is a more stark difference. This should be included as additional data if possible.

Validity of the findings

Data are robust and sound. The authors did a good job at reporting their results. Conclusions are mostly well stated and the paper is referenced well. The authors attempt to extrapolate their data to the role of autotaxin which I find inadvisable. They have found a potentially interesting biomarker, and that is what I would focus on.

---

## Round 0.2 · accepted · Accept

We are delighted to inform you that the manuscript was accepted for publishing at PeerJ. Thank you for contributing such a great job to PeerJ.

·

Basic reporting

I have no additional comments.

Experimental design

I have no additional comments.

Validity of the findings

I have no additional comments.

Additional comments

I have no additional comments.